# A Chinese–Kazakh Translation Method That Combines Data Augmentation and R-Drop Regularization

Canglan Liu [1,2,3], Wushouer Silamu [1,2,3] and Yanbing Li [1,2,3,*]

1 College of Computer Science and Technology, Xinjiang University, No. 777 Huarui Street, Urumqi 830017, China; 107552103739@stu.xju.edu.cn (C.L.); wushour@xju.edu.cn (W.S.)
2 Xinjiang Laboratory of Multi-Language Information Technology, Xinjiang University, No. 777 Huarui Street, Urumqi 830017, China
3 Xinjiang Multilingual Information Technology Research Center, Xinjiang University, No. 777 Huarui Street, Urumqi 830017, China
* Correspondence: liyb@xju.edu.cn

**Abstract:** Low-resource languages often face the problem of insufficient data, which leads to poor quality in machine translation. One approach to address this issue is data augmentation. Data augmentation involves creating new data by transforming existing data through methods such as flipping, cropping, rotating, and adding noise. Traditionally, pseudo-parallel corpora are generated by randomly replacing words in low-resource language machine translation. However, this method can introduce ambiguity, as the same word may have different meanings in different contexts. This study proposes a new approach for low-resource language machine translation, which involves generating pseudo-parallel corpora by replacing phrases. The performance of this approach is compared with other data augmentation methods, and it is observed that combining it with other data augmentation methods further improves performance. To enhance the robustness of the model, R-Drop regularization is also used. R-Drop is an effective method for improving the quality of machine translation. The proposed method was tested on Chinese–Kazakh (Arabic script) translation tasks, resulting in performance improvements of 4.99 and 7.7 for Chinese-to-Kazakh and Kazakh-to-Chinese translations, respectively. By combining the generation of pseudo-parallel corpora through phrase replacement with the application of R-Drop regularization, there is a significant advancement in machine translation performance for low-resource languages.

**Keywords:** machine translation; data augmentation; phrase replacement; R-Drop

## 1. Introduction

Neural Machine Translation (NMT) is a method of machine translation that uses neural networks [1–5]. NMT has achieved excellent performance on some resource-rich corpora. However, most languages in the world are low-resource, meaning they lack sufficient parallel data for training NMT models. This makes translation for low-resource languages challenging. Improving the performance of low-resource machine translation is currently a research focus [6,7].

Researchers have proposed various methods to improve the performance of low-resource machine translation. One method is transfer learning. Transfer learning improves the translation performance by transferring knowledge from a large-scale, high-resource corpus to a low-resource corpus [8–12]. This method can leverage the features and model parameters of the existing data for pre-training and fine-tuning in low-resource conditions to better adapt to the characteristics of the low-resource corpus. However, transfer learning requires more computational resources. Another method is to generate pseudo-parallel data using data augmentation techniques. Under low-resource conditions, this method can help improve machine translation performance. Data augmentation methods include

a variety of techniques, such as synonym replacement, sentence rearrangement, word insertion and deletion, and back-translation [13–18]. By leveraging these techniques, the original dataset can be expanded to provide more training samples, thereby improving machine translation performance under low-resource conditions. However, data augmentation techniques have some limitations [19], such as the possibility of generating unnatural translations, which may affect the accuracy of the translation.

Synonym replacement involves replacing certain words in the original sentence with their synonyms to generate new sentence pairs, thereby increasing data diversity [13,14]. Sentence shuffling rearranges the word order in the original sentence to create sentence pairs with the same semantics but different structures, helping the model learn different sentence structures and grammar rules. Word insertion and deletion can insert or remove certain words in the original sentence, generating slightly modified sentence pairs to increase data richness and diversity [15]. Back-translation involves first training an intermediate system with parallel data and then using that system to translate monolingual data in the target language back into the source language [16–18].

Although back-translation has been widely used in data augmentation, it may encounter issues such as translation errors and fluency due to the limitations of the machine translation model [20]. These problems can lead to significant differences between the generated pseudo-parallel data and real parallel data. Our plan is to augment parallel corpora using phrase replacement and increase data diversity, while reducing noise through various data augmentation methods. Additionally, we introduce the R-Drop regularization method to improve the robustness of the model [21]. With these measures, we aim to train an outstanding Chinese–Kazakh machine translation model.

The contributions of this work can be summarized as follows:

1. Parallel corpus augmentation: We successfully expanded the parallel corpus required for the Chinese–Kazakh machine translation task by utilizing phrase replacement techniques. By introducing more variations and diversities, we increased the richness of the training data, providing more information and context for training machine translation models.

2. Joint data augmentation: To improve the quality of low-resource language machine translation, we adopted a combination of various data augmentation methods. In addition to generating pseudo-parallel corpora through phrase replacement, other data augmentation methods, such as random phrase replacement and deletion flipping, were also employed. By combining different data augmentation methods, we further increased the diversity of the training data, enhancing the model's adaptability to various scenarios.

3. Introducing the R-Drop regularization method: By introducing the R-Drop regularization method, we effectively enhanced the robustness of the model. R-Drop ensures consistency between the outputs of two sub-models by minimizing the bidirectional Kullback–Leibler (KL) divergence. During training, R-Drop regularizes the outputs of two sub-models randomly sampled from dropout. This alleviates the inconsistency between the training and inference stages, strengthening the model's generalization ability and adaptability to unknown data.

The structure of this paper is as follows: the second part introduces the related work, the third part introduces the method proposed by us, the fourth part mainly introduces the experiments we conducted, as well as some experimental details, and finally, there are analyses and conclusions.

## 2. Related Work

Neural Machine Translation (NMT) is a method that uses neural network models to achieve automatic translation. Among them, Transformer is a classical NMT model that has achieved breakthrough results in the field of machine translation. Before Transformer, traditional machine translation methods were mainly based on Statistical Machine Translation (SMT) [22,23]. These methods often rely on phrase tables and language models, requir-

ing preprocessing of large-scale parallel corpora. However, SMT methods face challenges in word order and modeling long-distance dependencies, resulting in limited translation quality. To address these issues, Neural Machine Translation emerged. All the research in this work is based on the Transformer model [3]. In the field of Neural Machine Translation, the Transformer model has been widely applied and achieved significant results. Early NMT models used Recurrent Neural Networks (RNN) as the main component, such as the Seq2Seq model based on an encoder–decoder framework [24,25]. However, RNNs have limitations in terms of difficult parallel computation and modeling long dependencies. To overcome these limitations, the Transformer model was introduced. The Transformer model is based on attention mechanisms, capturing contextual information in input sentences through self-attention and multi-head attention mechanisms [26,27]. It consists of an encoder and a decoder, where the encoder transforms the source language sentence into continuous vector representations, and the decoder generates the target language sentence based on the output of the encoder. The introduction of the Transformer model has had a significant impact on the field of machine translation. By using self-attention mechanisms, the Transformer can better capture long-distance dependencies in input sentences, thereby improving translation quality. Additionally, the Transformer model has good parallel computing performance, making the training process more efficient.

In machine translation data augmentation, data are one of the key elements for training models. However, data scarcity has been a long-standing concern due to the challenges and cost constraints of acquiring high-quality parallel corpora [28,29]. To address this problem, researchers have proposed various methods for machine translation data augmentation. One standard data augmentation method is to generate synthetic data to expand the training set. For example, rule-based methods can generate new parallel sentence pairs through operations such as synonym substitution, deletion, and insertion. These synthetic data can enrich the training set and improve the generalization ability of machine translation models. Another data augmentation method is to leverage unsupervised learning with a large amount of monolingual data [30–32]. Techniques such as autoencoders and language models can be used to learn helpful language representations from monolingual corpora. These learned representations can initialize or fine-tune machine translation models, improving performance.

Some studies combine the original phrase table with parallel corpora for data augmentation [33,34]. However, this did not improve the machine translation system due to the high number of noisy and repetitive phrase pairs in the original phrase table. Researchers have tried to extract the longest unique phrase pairs from the phrase table and filter them using LaBSE to obtain high-quality phrase pairs [35]. Unfortunately, we could not use this method, as LaBSE does not support the Xinjiang Kazakh language.

In this study, we introduced a simple yet effective data augmentation method–phrase replacement technique, to address the issue of data scarcity in monolingual data and achieved significant performance improvement. This method is rule-based and generates new sentences by replacing phrases in the source language sentence. Specifically, we first use predefined rules and heuristics to identify and extract phrases from the source language sentence. Then, based on predefined replacement rules, these phrases are replaced with other phrases that have similar meanings or are contextually relevant. Finally, pseudo-parallel data is constructed based on the replaced source language sentence and the original target language sentence.

R-Drop is a consistency training method based on dropout, aiming to improve the robustness and generalization ability of a model. By minimizing the bidirectional KL divergence between the output distributions of sub-models sampled from dropout, R-Drop makes the model more stable to variations in input data. Traditional dropout randomly sets neurons to zero during training to reduce the model's reliance on certain features, thereby improving its robustness. However, dropout does not explicitly specify which features are important, which can lead to significant differences in output distributions between different sub-models. R-Drop addresses this issue by introducing a consistency

loss. Specifically, R-Drop samples two sub-models from the dropout and calculates the KL divergence between their output distributions. By minimizing this KL divergence, R-Drop forces the output distributions of the two sub-models to be as close as possible, enabling the model to better adapt to subtle variations in the input [21]. Through this consistency training approach, R-Drop encourages the model to learn more robust representations and generalize better to unseen examples. It promotes stability by reducing the discrepancy between sub-models and enhances the model's ability to handle input variations effectively.

## 3. Method

As shown in Figure 1, we first use the Moses system to generate an aligned phrase table from the fundamental parallel corpus (Bitext). Then, we prune the phrase table to remove low-quality or weakly aligned phrase pairs and filter out phrase pairs with low semantic information based on their part-of-speech tags. Next, we use the existing phrase table to replace phrases in the Bitext to generate pseudo-parallel data. After the data preparation stage, we merged the Bitext-generated pseudo-parallel data and trained the model. Finally, we fine-tune the model on the fundamental parallel corpus using the R-Drop regularization method to improve the model's performance further.

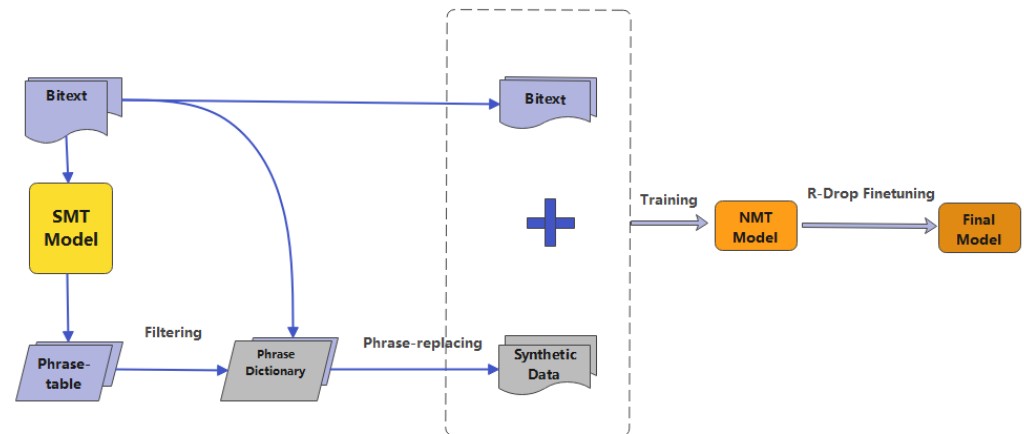

**Figure 1.** Illustration of our proposed method.

### 3.1. Generating a High-Quality Phrase Table

The Moses system is a phrase-based statistical machine translation system that utilizes a phrase table to establish translation correspondences between the source language and the target language [36]. The translation process in Moses is conducted through a decoding procedure. The phrase table stores phrase pairs and their translation probabilities, learned from parallel corpora. In the Moses system, the phrase table is considered one of the core components. It is a data structure that stores the correspondence between source language phrases and target language phrases. The phrase table contains phrase pairs and their associated translation probabilities, which indicate the likelihood of translation between the source and target phrases. These probabilities are estimated based on the training data set and statistical algorithms, often using maximum likelihood estimation. By utilizing the information stored in the phrase table during the decoding process, the Moses system can select the most appropriate translations for the input source sentences. The translation probabilities in the phrase table play a crucial role in determining the quality and accuracy of the translation output. Continuous refinement and augmentation of the phrase table through additional training data can improve the performance of the Moses system. Each phrase pair has five features:

- The phrase translation probability $\varphi(f|e)$;
- The lexical weighting $lex(f|e)$;
- The phrase inverse translation probability $\varphi(e|f)$;

- The inverse lexical weighting lex(e | f);
- The phrase penalty, currently always e = 2.718.

The first four features have probabilistic values ranging from zero to one, while the fifth feature remains constant.

In order to reduce noise in the phrase table, pruning is applied. When performing phrase table pruning, a threshold-based method can be used. The threshold is a predefined value that retains only those phrase pairs with probabilities higher than the threshold while removing phrase pairs with probabilities lower than the threshold. The benefit of using a threshold for phrase table pruning is filtering out phrase pairs with very low probabilities, as these are often unlikely to occur or irrelevant to the task. By removing these less important phrase pairs, the size of the phrase table can be significantly reduced while improving its quality and effectiveness [37].

To further enhance the quality of the phrase table, we can use part-of-speech (POS) filtering rules for refinement. POS filtering means selectively retaining or removing phrases based on specific POS tags. Since there is no POS tagging tool for Xinjiang Kazakh, we use Jieba to assign POS tags to each word in the Chinese phrases. Then, based on specific requirements, we retain or remove phrases with certain POS tags. We judge the POS tag of the last word in the Chinese phrase. Common POS tags include nouns (NN), verbs (VB), adjectives (JJ), and adverbs (RB). These POS tags often contain important semantic information and are thus retained in the phrases. On the other hand, POS tags such as determiners (DT) (such as "some", "every", and "any"), pronouns (PRP) (such as "you", "I", "she"), and prepositions (IN) often carry little semantic information but are abundant in the phrase table. By deleting these low-semantic Chinese phrases and their corresponding Kazakh phrases, we can effectively reduce redundancy and improve the quality of the phrase table. Since many irrelevant phrase pairs are deleted, the remaining phrase pairs are more similar in structure, reducing the noise introduced by the pseudo data generated by phrase substitution.

### 3.2. Generating Pseudo-Parallel Data Using a Phrase Table

Research has shown that retaining only unique phrase pairs from the extracted phrase table is important, as there may be multiple duplicate phrase pairs within the table. Previous methods often directly utilized phrase pairs generated by Moses as pseudo-parallel data, but this approach may not yield satisfactory results for Chinese and Kazakh translations.

First, we use a phrase dictionary to store the mapping between Chinese and Kazakh phrases. Then, we read a bilingual corpus containing sentences in both Chinese and Kazakh. In the main loop of the code, we iterate through each sentence and look for matching pairs of Chinese and Kazakh phrases within each sentence. Once a matching phrase pair is found, we randomly select a Kazakh phrase from the phrase dictionary as a replacement and substitute the current target Kazakh phrase in the sentence. At the same time, we replace the Kazakh phrase with the corresponding Chinese phrase based on the phrase dictionary.

Finally, we consider the replaced Chinese sentence and the substituted Kazakh sentence as pseudo-parallel data.

### 3.3. Utilizing R-Drop for Model Fine-Tuning

To overcome the problem of overfitting when using large models on small datasets, we introduced the Dropout technique for regularizing training of deep neural networks [38]. Although Dropout is very effective, it introduces randomness, leading to noticeable inconsistency between training and inference. Therefore, we adopted the R-Drop training strategy, which aims to mitigate the inconsistency between training and inference by making the output distributions of different sub-models generated by Dropout consistent with each other. Specifically, in each mini-batch, we perform two forward passes for each data sample. R-Drop constrains the different sub-models generated by Dropout by minimizing the bidirectional KL divergence to encourage their output distributions to be as consistent as possible. In this way,

R-Drop is able to better balance the output differences between different sub-models and improve the consistency of the model between the training and inference stages.

Specifically, when given training data $D = \{x_i, y_i\}_{i=1}^n$, for each training sample $x_i$, it undergoes two forward propagations through the network, resulting in two output predictions: $P_1(y_i|x_i)$ and $P_2(y_i|x_i)$. Due to Dropout randomly dropping some neurons each time, $P_1$ and $P_2$ are different prediction probabilities obtained from two different subnetworks (from the same model) (as shown in Figure 2). R-Drop leverages the difference between these two prediction probabilities and applies symmetric Kullback–Leibler (KL) divergence to constrain $P_1$ and $P_2$ :

$$L_{KL}^i = \frac{1}{2}(D_{KL}(P_1(y_i \mid x_i) \| P_2(y_i \mid x_i)) + D_{KL}(P_2(y_i \mid x_i) \| P_1(y_i \mid x_i))) \tag{1}$$

Additionally, the conventional maximum likelihood loss function is included.

$$L_{NLL}^i = -\log P_1(y_i \mid x_i) - \log P_2(y_i \mid x_i) \tag{2}$$

The final training loss function is:

$$L_i = L_{NLL}^i + \alpha * L_{KL}^i. \tag{3}$$

Here, $\alpha$ is used to control the coefficient of $L_{KL}^i$, making the training of the entire model straightforward. R-Drop, on the other hand, constrains the parameter space by deliberately imposing constraints on the outputs between sub-models during the training process, ensuring consistency among different outputs and reducing the inconsistency between training and testing.

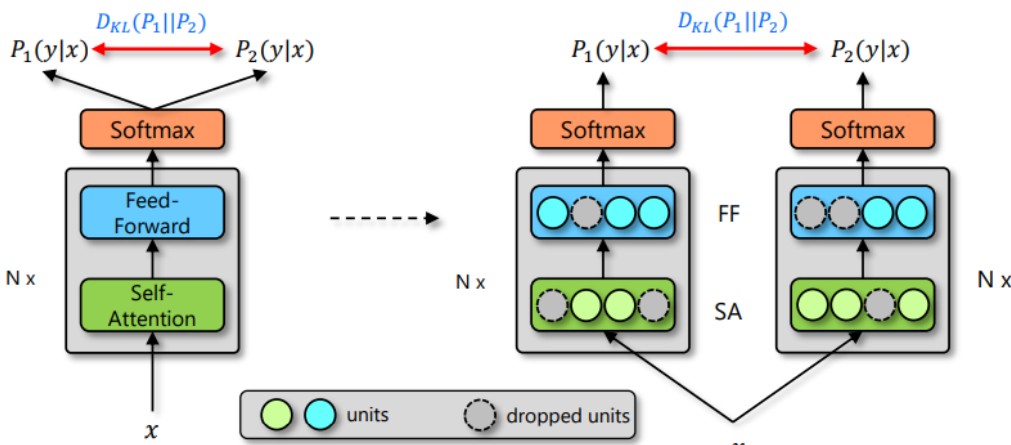

**Figure 2.** The overall framework of R-Drop.

By adopting the R-Drop training strategy, we can alleviate the issue of inconsistency between training and inference and enhance the generalization ability of deep neural networks on small datasets. This approach helps optimize the performance and stability of the model, thereby improving its effectiveness in practical applications.

## 4. Experiments

### 4.1. Data and Preprocessing

In terms of experimental design, the study aims to conduct machine translation experiments from Chinese to Kazakh. A dataset of 30.4k bilingual sentences collected by the laboratory is used, with 2k sentences for validation and 2k sentences for testing. In data preprocessing, several steps were applied to the training set. Firstly, sentences longer than 200 tokens were removed to ensure data integrity and reduce noise. Secondly, sentences with a length ratio greater than three were discarded to avoid extremely long or short text samples. Additionally, punctuation normalization and uniform handling of half-width

and full-width characters were performed to improve data consistency and processability. For Chinese text segmentation, the jieba segmentation tool was used to tokenize Chinese sentences into word units, facilitating better handling of semantic information [39]. For Kazakh text segmentation, the Moses tokenization tool was chosen to provide standardized input data.

### 4.2. System Environment and Model Parameters

In our experiment, we used Ubuntu 20.04.6 as the operating system environment and trained the model on a Tesla V100 GPU. The system had 16 GB of memory capacity. To build a shared vocabulary, we utilized the subword-nmt tool and set the vocabulary size to 10k to ensure consistency in word representation between the source and target languages [40,41]. We chose the fairseq framework as the underlying framework, which provides rich functionalities and efficient operations, supporting model training and execution effectively with high computational performance [42].

Here are the detailed descriptions of the model parameters: the model architecture is the Transformer, the label smoothing coefficient is set to 0.1, the dropout rate is 0.1, the weight decay coefficient is 0.01, and the dropout rate of the attention mechanism is 0.1. The Adam optimizer is used for parameter update, and the beta parameter of the Adam optimizer is set to 0.9 and 0.98 [43]. The first 4000 update steps use linear learning rate preheating. The initial learning rate in the preheating stage is $1 \times 10^{-7}$, the basic learning rate during training is 0.0003, and the maximum number of tokens per training batch is 4096. The model parameters were not manually or automatically tuned. sacreBLEU is a popular tool for computing BLEU (Bilingual Evaluation Understudy) and chrF++ (chrF enhanced with word bigrams) scores, measuring the similarity between machine-translated output and reference translations [44–47]. BLEU is a commonly used metric to evaluate machine translations. It measures the similarity between machine-translated outputs and reference translations at the word level. The calculation formula for BLEU is as follows:

$$\text{BLEU} = BP \cdot \exp\left(\sum_{n=1}^{N} w_n \log p_n\right) \tag{4}$$

$$BP = \begin{cases} 1 \text{ if } c > r \\ e^{1-r/c} \text{ if } c \leq r \end{cases} \tag{5}$$

The variables $c$ and $r$ represent the length of the translation and reference text, respectively, to be evaluated. chrF++ is a metric that evaluates machine translation quality by combining both character-level and word-level matching and calculating their average value. It addresses the limitations of traditional metrics like BLEU when evaluating long sentences or sentences with more errors.

$$\text{chrF}\beta = \left(1 + \beta^2\right) \frac{\text{chr}P \cdot \text{chrR}}{\beta^2 \cdot \text{chr}P + \text{chr}R} \tag{6}$$

In this context, the default value of $\beta$ is 2.chrP refers to Precision, which measures the proportion of character-level n-grams that match the candidate and reference translations. chrR represents Recall, which measures the proportion of character-level n-grams matched between the candidate and reference translations.

### 4.3. Results and Discussion

4.3.1. Baseline

It is logical to compare our proposed method with similar approaches. Thus, we have chosen highly similar approaches for comparison as follows:

Transformer: For our experimental section, we opted to use the Transformer without any additional enhancement methods as our foundation model [3]. This benchmark model was trained and tested on raw data to assess the effectiveness of other enhancement meth-

ods. With this strong baseline model, we can accurately gauge the impact of other methods on model performance and implement specific improvements.

Back-translation: In the field of Neural Machine Translation (NMT), the technique of Back-translation involves the utilization of a target-to-source translation model to create artificial source sentences [16]. These synthetic sentences are subsequently incorporated into the original source-to-target translation model as supplementary training data, thereby enhancing its effectiveness.

Replace: In this approach, a portion of the words aligned between the source language and target language are replaced. The authors use a bilingual lexicon obtained from the training corpus to randomly select $\alpha$ times the number of aligned words and replace them with random entries from the lexicon [48]. The purpose of this is to introduce vocabulary that is difficult to generate solely based on the target language prefix, thereby forcing the system to pay attention to the words in the source language.

Token: When generating new words, a technique called Token: $\alpha \cdot t$ replaces some target words with a special (UNK) token [49]. This is performed to make the target prefix less informative, which pushes the system to rely more on the encoder. It is similar to word dropout, which prevents posterior collapse in variational autoencoders.

Swap: During the Swap task, pairs of randomly selected target words are swapped until only $(1 - \alpha) \cdot t$ words remain in their original position [50]. This helps the system generate new words by relying less on the target prefix.

Source: The most efficient way to generate correct output is by copying directly from the source sentence. However, some researchers have identified this approach as detrimental to Neural Machine Translation (NMT) [51]. Studies indicate potential drawbacks, while others have demonstrated the usefulness of only copying in the inverse direction.

Reverse: The encoder's influence decreases when the target sentence's word order is reversed. Reversing the order improves the system's ability to use encoder information when generating end-of-sentence words [52].

The hyperparameter $\alpha$ controls the proportion of words affected by swap, token, and replace transformations. Its default value is 0.1.

### 4.3.2. Result

Table 1 presents the impact of various data augmentation methods on translation performance, measured in terms of BLEU and chrF++ scores. The following are the experimental results.

**Table 1.** Performance of several data augmentation methods on Zh–Kk and Kk–Zh translation tasks with the SacreBLEU metric based on the transformer model.

| | Zh–Kk | | Kk–Zh | |
|---|---|---|---|---|
| | BLEU | chrF++ | BLEU | chrF++ |
| Transformer | 49.47 | 0.745 | 52.04 | 0.463 |
| Back-translation | 49.93 | 0.746 | 54.21 | 0.478 |
| Replace | 49.90 | 0.747 | 57.15 | 0.514 |
| Token | 49.26 | 0.744 | 56.99 | 0.512 |
| Swap | 48.91 | 0.742 | 57.15 | 0.513 |
| Source | 48.93 | 0.742 | 52.14 | 0.462 |
| Reverce | 49.81 | 0.750 | 57.85 | 0.516 |
| Phrase-substitution | 50.15 | 0.752 | 57.35 | 0.514 |

For the Chinese to Kazakh translation task, we observed a BLEU score of 49.47 and a chrF++ score of 0.75 using the Transformer baseline model. However, when the phrase replacement data augmentation method was applied, we observed an improvement in the

BLEU score to 50.15 and the chrF++ score to 0.76. This indicates that the phrase replacement data augmentation method has a significant performance improvement effect on the low-resource translation task from Chinese to Kazakh. Additionally, we observed that methods such as Token, Swap, and Source not only did not enhance the performance, but also impaired the translation quality. By applying the phrase replacement data augmentation method, more training samples can be generated, thereby increasing the amount and diversity of the training data for the model. More training data provides comprehensive information, enabling the model to learn better language representations and translation capabilities.

For the translation task from Kazakh to Chinese, under the baseline Transformer model, it was observed that the BLEU score was 52.04 and the chrF++ score was 0.46. However, by applying the phrase replacement data augmentation method, it was observed that the BLEU score improved to 57.35, and the chrF++ score improved to 0.51. This once again demonstrates the effectiveness of the phrase replacement data augmentation method in low-resource translation tasks from Kazakh to Chinese. Furthermore, it was also observed that using the Reverse augmentation method resulted in a BLEU score increase to 57.85. Compared to other data augmentation methods, Reverse introduces more diverse language structures and contextual environments by reversing the Kazakh sentences. This helps the model better learn the correspondence between the source language and the target language, thereby improving translation accuracy and naturalness. Meanwhile, the Source augmentation method did not significantly improve machine translation performance for languages such as Chinese and Kazakh, which have rich morphological characteristics. Based on the above experimental results, it is recommended to consider using data augmentation methods such as phrase replacement and Reverse to further improve translation quality in the task of translating from Kazakh to Chinese. Additionally, depending on the specific task and data conditions, other data augmentation methods can be explored to discover better performance enhancement strategies.

### 4.3.3. Combining Multiple Augmentation Methods

To further improve the machine translation performance of the Kazakh language, we conducted a combination experiment using multiple data augmentation methods. We selected four commonly used data augmentation methods: Phrase-substitution, Reverse, Swap, and Token for comparison. Here are our experimental content and results.

Firstly, we combined the Phrase-substitution method with the Reverse method and used it as the base model for mixed data augmentation. The experimental results showed that compared to using only the Phrase-substitution method, this combination strategy significantly improved translation quality, demonstrating higher BLEU scores.

We extended the Phrase-substitution + Reverse method and combined it with the Token and Swap methods. The experimental results showed that in the Chinese-to-Kazakh translation task, the Phrase-substitution + Reverse + Swap method achieved the highest BLEU score, indicating better performance improvement. In contrast, the Phrase-substitution + Reverse + Token method showed a slight improvement in both tasks, but the improvement was not significant. However, in the Kazakh-to-Chinese translation task, the Phrase-substitution + Reverse + Swap method performed poorly, even lower than the performance of the base model.

Overall, the effectiveness of data augmentation methods in machine translation tasks depends on the specific language pair and task type. The Phrase-substitution + Reverse + Swap method performed relatively well in the Chinese-to-Kazakh translation task, while the Phrase-substitution + Reverse + Token method performed better in the Kazakh-to-Chinese translation task. The experimental results are shown in Table 2.

**Table 2.** Performance of data augmentation methods on Zh–Kk and Kk–Zh tasks.

| | Zh–Kk | | Kk–Zh | |
|---|---|---|---|---|
| | BLEU | chrF++ | BLEU | chrF++ |
| Phrase-sub.+Rev. | 50.42 | 0.756 | 58.74 | 0.530 |
| Phrase-sub.+Rev.+Token | 50.55 | 0.756 | 58.99 | 0.532 |
| Phrase-sub.+Rev.+Swap | 51.46 | 0.761 | 58.58 | 0.527 |

4.3.4. Fine-Tuning Using R-Drop

In low-resource environments, models may be affected by noise and overfitting. The R-Drop regularization method was introduced to reduce overfitting in Chinese–Kazakh low-resource machine translation. We pretrain the model with mixed data augmentation training and fine-tune it using real parallel corpora. Different $\alpha$ values were tested to find the optimal hyperparameter settings. Experimental results are shown in Table 3.

**Table 3.** Performance with different $\alpha$ values for fine-tuning on the Zh–Kk and Kk–Zh tasks.

| | Zh–Kk | | | Kk–Zh | | |
|---|---|---|---|---|---|---|
| $\alpha$ | BLEU | chrF++ | Time | BLEU | chrF++ | Time |
| 0.3 | 53.56 | 0.771 | 3.11h | 59.53 | 0.539 | 4.49h |
| 0.4 | 54.46 | 0.777 | 4.48h | 60.15 | 0.544 | 5.46h |
| 0.5 | 54.45 | 0.776 | 5.35h | 59.59 | 0.538 | 3.35h |
| 0.6 | 54.29 | 0.774 | 5.96h | 59.28 | 0.535 | 3.05h |
| 0.7 | 54.44 | 0.777 | 6.38h | 59.74 | 0.539 | 5.64h |

The table shows that increasing the KL divergence weight parameter improves translation performance for both Chinese-to-Kazakh (Zh–Kk) and Kazakh-to-Chinese (Kk–Zh) tasks, as evidenced by higher BLEU and chrF++ scores. The optimal parameter value for peak performance in both tasks is 0.4, according to the provided data. However, a trade-off exists between translation performance and time consumption. Increasing the parameter value results in longer training time, with weight 0.7 taking twice as long as weight 0.3 when fine-tuning from Chinese to Kazakh. The time required for Kazakh to Chinese also increases with the weight.

During training, the KL divergence weight parameter controls the KLDivLoss weight and affects the final loss function. Larger values introduce a stricter contrastive learning objective, forcing the model to better adjust the distribution difference between the two logits but may require more training steps to converge. Therefore, choosing the appropriate parameter value requires careful consideration of both translation performance and time consumption.

However, a trade-off exists between translation performance and time consumption. Increasing the parameter value results in longer training time, with weight 0.7 taking twice as long as weight 0.3 when fine-tuning from Chinese to Kazakh. The time required for Kazakh to Chinese also increases with the weight.

*4.4. Qualitative Analysis*

We randomly selected Chinese to Kazakh translation models and present a qualitative analysis in Table 4. It includes the source sentence, the reference translation, and the machine-generated translations from different NMT models.

**Table 4.** An example of using the model trained with methods such as Transformer, Phrase Substitution, Mixed Enhancement (Phrase-replacement + Reverse + Swap), and Mixed Enhancement + R-Drop (with Chinese as the source language and Kazakh as the target language).

| Method | Translations from Chinese to Kazakh |
| --- | --- |
| Source Sentence | 认真解决群众反映强烈的商业促销噪音污染、占道经营阻碍交通等信访投诉问题。 (Seriously address the strong public complaints regarding commercial promotion noise pollution, obstruction of traffic due to road occupation, and other petition issues.) |
| Reference | كۇشتى سىسىتى ناعلوب ادؤاس ىلىقرا ساتؤەساس جەھەبۇ جۇىلەۋ شواتساس ، ، كۇبۇقاراني ىن شاعىمى - ارىز ، ارمان - ارىز سىياقتى قاتىناس جۇرگىزؤزى جارات ىت پالى ارات ىي ەلەھپ جولدى ن ەرلەھلەس ماس تاىيقۇم ش ش. كەرەك) (Seriously address the strong public complaints regarding commercial promotion noise pollution, obstruction of traffic due to road occupation, and other petition issues.) |
| Baseline | ش ؤىلدى كۇ جەھەبۇ ساتؤەساس ارقى ىلى ساؤدا بولعان كۇشتى سىسى ى كۇبۇقاراني ىن قاتىناس ى ساناس ۇشى ىراۋ كە ەدەرگى ى گە جۇرگى ىزؤ ت ىي جارات ىي ەلەھپ جولدى ، لاس تانۇى ارى ز سى ياقتى. كەرەك ش ش مؤقىيات ماس ەلەس ى ن ايت ۇ ارمان - ارى ز سى ياقتى (Seriously address the issues raised by the masses regarding problems such as noise pollution caused by commercial promotions, obstructed traffic due to business operations occupying roads, and other petitions.) |
| Phrase-substitution | ش ؤىلدى كۇ جەھەبۇ ساتؤەساس ارقى ىلى ساؤدا بولعان كۇشتى سىسى ى كۇبۇقاراني ىن ارى ز سى ياقتى قاتىناس بولاتى ىن كە ەدەرگى ى ت ىي جارات ىي اقتى ەلەھي ت ىي ن جول ، لاس تانۇى - ارى ز ، ارمان - ارى ز ش اعى ن ەرلەھلەس ماس تاىيقؤم ش ش. كەرەك) (Seriously address the petitions and complaints raised by the public regarding the strong impact of noise pollution caused by commercial promotions and the obstruction of traffic due to business operations occupying public spaces.) |
| Mixed Enhancement | كۇبۇقاراني ىن كۇشتى سىسى ى بولعان ساؤدا ساتؤەساس جەھەبۇ كۇ ش ؤىلدى ، لاس تانۇ جولدى ارمان - ارى ز سى ياقتى قاتىناس بولاتى ىن كە ەدەرگى ى جۇرگى ىزؤ ىنە ت ىي جارات ىي ەلەھپ ن ەرلەھلەس ماس تاىيقؤم ش ش. كەرەك) (Seriously address the petitions and complaints raised by the public regarding the strong impact of noise pollution caused by sales promotions and the obstruction of traffic due to business operations occupying roadways.) |
| Mixed Enhancement+R-Drop | كۇبۇقاراني ىن الۇ ش ش سىسى ى كۇشتى ناعلوب ق ىي دالاؤدا ساتؤەساس جەھەبۇ كۇ ش ؤىلدى ناس تانۇى ، جولدى ەلەھي ت ىي جارات ىي نە ؤىگى جۇرگ كە ەدەرگى ى ن بولاتى ىن قاتى ناس سى ياقتى ارى ز - ارمان ، ارى ز - ش اعى ن ەرلەھلەس ماس تاىيقؤم ش ش. كەرەك) (Seriously address the petitions and complaints raised by the public regarding the strong impact of noise pollution caused by commercial sales promotions and the obstruction of traffic due to businesses occupying roadways.) |

Using the table, we can compare the reference sentence with the translations generated by the NMT models. The baseline translation is generally similar to the reference translation but contains some vocabulary choices and translation errors. For example, "ەلەھپ جولدى ي (obstructed traffic due ارات ىي جارات ىت گە ەدەرگى ى ۇشى ىراۋ ساناس ى قاتى ناس سى ياقتى ارى ز" to business operations occupying roads) should be "ەلەھپ الپى ي ارات ىت جارات ىي ؤزؤ جۇرگى ىزؤ جولدى" ساناس ى قاتى ناس سى ياقتى ارى ز ، ارمان ، ؤ (obstruction of traffic due to road occupation); some translation errors need to be corrected. The phrase-substitution method improves the baseline translation by fixing vocabulary choices and translation errors. It is closer to the expression of the source sentence and better conveys the original meaning. The mixed enhancement method further improves the translation to make it more accurate and fluent. It is closer to the source sentence regarding grammar, structure, and vocabulary choices while maintaining a certain natural fluency. The mixed enhancement+R-Drop method further simplifies the translation by omitting redundant words, making it more concise.

Overall, the reference translation, phrase-substitution, mixed enhancement, and mixed enhancement + R-Drop translation methods improve the source sentence to varying degrees



and strive to convey the original meaning. The mixed enhancement + R-Drop method may be the optimal choice, as it balances accuracy and conciseness.

Table 5 shows a Kazakh source sentence and its corresponding Chinese reference translation. The table also includes machine-translated Chinese sentences generated by different NMT models. Notably, using the Mixed Enhancement+R-Drop approach produced more outstanding translations.

**Table 5.** An example of using the model trained with methods such as Transformer, Phrase Substitution, Mixed Enhancement (Phrase-replacement + Reverse + Token), and Mixed Enhancement + R-Drop (with Kazakh as the source language and Chinese as the target language).

| Method | Translations from Kazakh to Chinese |
| --- | --- |
| Source Sentence | ، لاستاۋ شۇيلىى جەبەۇ ساتۇدى ارقىلى ساۇدا بولعان كۇشتى الىسى بۇڭارانىڭ شاعىم - ارىز ، ارمان - ارىز سياقتى قاتىناس جۇرگىزۇ تيجارات الىپ يەلەپ جولدى (Seriously address the strong public complaints regarding commercial promotion noise pollution, obstruction of traffic due to road occupation, and other petition issues.) |
| Reference | 认真解决群众反映强烈的商业促销噪音污染、占道经营阻碍交通等信访投诉问题。(Seriously address the strong public complaints regarding commercial promotion noise pollution, obstruction of traffic due to road occupation, and other petition issues.) |
| Baseline | 要认真解决群众反映强烈的贸易促进噪声污染、道路占用经营交通等信访问题。(Address the trade promotion noise pollution, road occupation, business traffic, and other petition issues that have been strongly complained by the public.) |
| Phrase-substitution | 要认真解决群众反映强烈的以商促销噪音污染、占道经营交通等诉求、诉讼问题。(Seriously address the strong public concerns regarding issues such as noise pollution caused by promotional activities conducted through commerce, traffic congestion due to roadside business operations, and related demands and litigation problems.) |
| Mixed Enhancement | 认真解决群众反映强烈的营商促销噪声污染、占路经营交通等信访投诉问题。(Seriously address the public complaints regarding noise pollution caused by commercial promotion activities, road occupation for business operations, and other petition issues.) |
| Mixed Enhancement+R-Drop | 认真解决群众反映强烈的以商促销噪声污染、占道经营交通等信访投诉问题。(Seriously address the public complaints regarding noise pollution caused by commercial promotions, road occupation for business operations, and other petition issues.) |

The baseline translation has made simple modifications to the original sentence, replacing some vocabulary but retaining the meaning. However, it uses inaccurate terms, such as translating "占道经营阻碍交通" (obstruction of traffic due to road occupation) as "道路占用经营交通" ( road occupation, business traffic). In the phrase "道路占用经营交通" (road occupation, business traffic), firstly, the verb "占用" (occupation) should come after the noun "道路" (road), resulting in "占用道路" (occupy the road). Therefore, the correct translation would be "occupation of the road". Secondly, considering the context, "经营交通" (business traffic) can be understood as traffic congestion caused by commercial activities. Hence, it should be translated as "obstruction of traffic due to road occupation".

Phrase substitution improves the accuracy and fluency of the translation by replacing some phrases. In this case, " 商业促销" (commercial promotion) is substituted with "以商促销" (promote through commerce). Mixed Enhancement of various modification techniques, including phrase substitution, vocabulary deletion, and word order adjustment, achieves a better translation result. In addition to phrase substitution, "商业" (business) is changed to "营商" (commercial), making the translation more contextually appropriate.

Mixed Enhancement + R-Drop Building upon mixed Enhancement: this method further omits some modifying words to simplify sentence structure and enhance conciseness. While this approach simplifies the translation, it may result in the loss of some detailed information.

In summary, the different translation methods vary in their degree of processing and effectiveness for the original sentence. Phrase-substitution, Mixed Enhancement, and

Mixed Enhancement + R-Drop improve the accuracy and fluency to a certain extent, while the baseline relatively contains some inaccuracies.

## 5. Analyses

Through the phrase replacement method, we have successfully improved the performance of low-resource machine translation. However, this method also has some limitations. Firstly, even after pruning and filtering, the remaining aligned phrase table is still relatively large, so generating pseudo-parallel corpora is time-consuming. Secondly, since the replacement is conducted randomly, the generated pseudo-parallel corpora may contain grammatical and semantic errors.

In future research, we plan to explore further the possibility of improving machine translation performance by replacing phrases with similar structures and introducing semantic constraints. Although our method still has some flaws, we have provided new ideas for machine translation data augmentation methods, and this method is relatively easy to implement in practical applications. Our research is meaningful to both the machine translation field and practical applications.

## 6. Conclusions

This paper proposes a method to augment the Chinese–Kazakh parallel dataset by phrase substitute. We compare our method with other data augmentation methods and show that it complements them. We also perform R-Drop fine-tuning experiments and try different KL divergence weight coefficients to improve the model performance further.

Our experimental results show that by combining data augmentation and R-Drop fine-tuning, we can achieve a significant improvement in BLEU score over the baseline Transformer model, with 4.99 and 7.7 improvements in Chinese-to-Kazakh and Kazakh-to-Chinese translation tasks, respectively. This suggests that our method is effective in improving translation quality.

However, it is essential to note that while our proposed data augmentation method is a feasible approach to improving NMT systems, especially in the case of low-resource or low-quality data, our use of random replacement may introduce noise into the generated pseudo-parallel data, as it does not consider semantic constraints. Therefore, future work could further explore ways to introduce semantic constraints or data filtering to improve the model performance.

In summary, our research demonstrates that by augmenting the dataset with phrase replacement and combining it with other techniques, we can improve the quality of Chinese-to-Kazakh and Kazakh-to-Chinese translations. These research findings are significant for promoting language translation and cross-cultural communication.

**Author Contributions:** C.L. designed the method and carried out the related experiments. C.L. wrote the manuscript. W.S. funded the research. Y.L. reviewed the manuscript. All authors have read and agreed to the published version of the manuscript.

**Funding:** This work was supported by the National Natural Science Foundation of China Joint Fund Project under Grants U1911401 and the National Natural Science Foundation of China Joint Fund Project under Grants U1603262.

**Institutional Review Board Statement:** Not applicable.

**Informed Consent Statement:** Not applicable.

**Data Availability Statement:** The experimental data was manually translated and proofread by the laboratory. If you wish to utilize the data for research purposes, you need to submit an application to the corresponding author.

**Conflicts of Interest:** The authors declared no potential conflict of interest concerning this article's research, authorship, and publication.

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
