# Peer review of "A Chinese–Kazakh Translation Method That Combines Data Augmentation and R-Drop Regularization"

_applsci, doi:10.3390/app131910589_

Round 1
Reviewer 1 Report
I was delighted to read this manuscript. However, I would like to suggest a few changes to improve its quality. I have stated my comments below. I hope the authors find them helpful.
Content
Data augmentation limitations: The authors wrote the following statement in Line 31, “data augmentation techniques have some limitations, such as the possibility of generating unnatural translations”. Although this seems evident, it would be useful to add a citation to a source which supports this statement. I appreciate the authors are experts in the field who do not need further sources, but some readers may benefit from having additional clarity on this statement. Ideally, the authors can illustrate the statement by referring to the work of other researchers or, at the very least, cite a source where this can be confirmed.
Back translation: Line 42 states the following, “Although back-translation has been widely used in data augmentation, it may encounter issues such as translation errors and fluency due to the limitations of the machine translation model”. Once again, the authors may not need further information about this, but readers may benefit from a citation to a source that confirms the authors’ point. Ideally, the authors can refer to an example or to the work of other researchers who can confirm this point about back translation.
Figure 1: The text makes no reference at all to Figure 1. There is no point in adding a figure to the manuscript if it is not going to be discussed. I realize the authors were trying to describe the Transformer in the first paragraph of the Related Work section, but the description does not correspond with the diagram displayed in Figure 1. For example, Line 90 talks about self-attention mechanisms, which are not depicted in Figure 1 (the diagram only includes multi-head mechanisms). Also, Figure 1 refers to “Add & Norm”, which we can guess it means “Add and normalize”, but this is not explained at all in the text around Figure 1 (the word normalization is mentioned only once in the manuscript, and this is well after Figure 1). If Figure 1 is not worth mentioning in the text, the authors must remove it. If Figure 1 is important, then the authors must describe it properly (do not make us guess why self-attention is not included or what norm means). If the text was taken from a source, then cite the source.
Figure 2: This is another figure which is not referred to in the text. If Figure 2 is not worth mentioning in the text, the authors must remove it. I would prefer to keep Figure 2, but it must be described properly. Do not assume that you can talk about Bitext in the diagram simply because it is a common term in machine translation. Explain in the text what you mean (this is the combination of both source and target language versions of a given text).
Table 4: I would strongly appreciate having an English translation for the text in Table 4, and I am sure many of your readers would appreciate it too. I realize that your work is not about English (Table 4 show Translations from Chinese to Kazakh, which is what you want to illustrate). However, the lack of English translations makes the text around Table 4 hard to follow. You have provided translations for some of the statements (Line 369), but not all. Add a third column to Table 4 with English translations (this is only to improve the readability of your text for people who cannot read the translations from Chinese to Kazakh).
Table 5: As in the case of Table 4, it would be useful to provide an English translation for the Translations from Kazakh to Chinese in Table 5.
English language and style
Contractions: Overall, the paper is easy to follow and pleasant to read. However, contractions must be avoided. Contractions are acceptable in informal writing situations, such as blog posts. However, this is an academic paper. Thus, replace It’s with It is in Line 278.
References (citations): Generally, the authors seem to add a full stop before entering a citation (except for the first paragraph). However, the sentences actually end after the citation (not before). Thus, the full stop should be shown after the citation. For example, in Line 23, “corpus.[8–12]” should be replaced with “corpus [8–12].” Note that the full stop appears after the citation (not before). Also, “back-translation.[13–18]” in Line 29 should be replaced with “back-translation [13–18].” Note that the full stop appears after the citation (not before). This should be corrected in the whole paper.
Caption for Table 1: Is there a hyphen missing between Zh and Kk, and then between Kk and Zh? At present, there is a small “rectangle” between these abbreviations which needs to be fixed.
Author Response
Thank you very much for your valuable comments and suggestions. These feedback are very helpful for improving the quality of the manuscript. I have highlighted the changes, and I will try my best to make the corresponding modifications and improvements in the final version. Please let me know if you have any other questions or comments.
Data augmentation limitations: You suggested adding a citation to support the statement that data augmentation can lead to unnatural translations. This is a good suggestion, as it can further support the statement by citing the work of other researchers. I have made the correction.
Back-translation: You suggested citing the work of other researchers to support the author's argument in the discussion of back-translation. This is also a good suggestion, as it can increase the reader's confidence in the author's argument by citing the results of other studies. I have made the correction.
Figures 1 and 2: You pointed out that Figures 1 and 2 are not adequately explained or cited in the text. You suggested that if Figures 1 and 2 are not important, they should be removed; if they are important, they need to be properly described and cited. This is also a reasonable suggestion. I have deleted Figure 1, and I have further explained Figure 2.
Tables 4 and 5: You suggested providing English translations for the Kazakh-to-Chinese and Chinese-to-Kazakh translations to improve readability. This is a good suggestion, as providing English translations for non-Chinese readers can help them better understand the content of the tables. I have added English translations.
English language quality assessment: You offered some suggestions on the use of punctuation in abbreviations and citations. It is important to use punctuation and abbreviations correctly in academic papers. I will incorporate these suggestions into the final revision.
Table 1 title: You pointed out that the hyphens are missing between "Zh" and "Kk" and between "Kk" and "Zh" in the title of Table 1. I will ensure that the correct hyphens are added in the revision to improve the appearance of the table.
Reviewer 2 Report
To the esteemed authors, I would like to express my heartfelt appreciation for your dedicated efforts and contributions and your selection of this reputed journal.
Abstract:
It is suggested to conclude the abstract with a concise one-sentence summary of the research to encapsulate its main findings.
Introduction:
To strengthen the credibility of the statement regarding the limitations of machine translation in back-translation, please provide a specific citation for the claim. Additionally, ensure that the citation for reference [19] accurately corresponds to the content it supports.
Consider including an outline of the paper at the conclusion of the introduction to offer readers a clear roadmap of the subsequent sections.
Related Work:
Correct the formatting so that references appear after the full stop and revise the structure to create distinct paragraphs. The same thing appears throughout all the paper.
Ensure that Figure 1 and Figure 2 is properly referenced within the text. (remove additional dot the figure caption)
Method:
Provide a rationale for the utilization of Part-of-Speech (POS) filtering in the phrase table refinement process. Specify what POS filtering entails, how it is implemented, and why it contributes to the enhancement of phrase table quality. For example, substantiate the statement, "Through POS filtering, we can eliminate low-information and frequently occurring but semantically insignificant phrases."
Experiments:
Include citations for tools used in the research, such as the "subword-nmt tool," to acknowledge their contribution.
Clarify whether the model parameters were manually tuned or if there was an automated optimization process.
For enhanced readability, consider providing English translations for examples, especially in qualitative analysis and tables (e.g., Table 4 and Table 5).
It is recommended to include a section that discusses the limitations of the research, potential threats to validity, and the practical implications of the study on both research and real-world applications.
These revisions and additions will contribute to the overall clarity, credibility, and comprehensibility of your paper.
Author Response
Thank you very much for your valuable comments and suggestions. These feedback are very helpful for improving the quality of the manuscript. I have highlighted the changes, and I will try my best to make the corresponding modifications and improvements in the final version. Please let me know if you have any other questions or comments.
Abstract:
I have revised the abstract to include a concise one-sentence summary of the research, encapsulating its main findings.
Introduction:
To strengthen the credibility of the statement regarding the limitations of machine translation in back-translation, I have added a specific citation to support the claim. Additionally, I have ensured that the citation for reference [19] accurately corresponds to the content it supports.
I have included an outline of the paper at the conclusion of the introduction to provide readers with a clear roadmap of the subsequent sections.
Related Work:
I have corrected the formatting so that references appear after the full stop, and I have revised the structure to create distinct paragraphs. This correction has been applied throughout the entire paper.
Based on the combined feedback from both editors, I have removed Figure 1 and provided further explanation for Figure 2 and I have removed any additional dots in the figure captions.
Method:
I have provided a rationale for the utilization of Part-of-Speech (POS) filtering in the phrase table refinement process. The explanation includes what POS filtering entails, how it is implemented, and why it contributes to enhancing the quality of the phrase table.
Experiments:
I have included citations for the tools used in the research, such as the "subword-nmt tool," to acknowledge their contribution.
I have clarified whether the model parameters were manually tuned or if there was an automated optimization process.
For enhanced readability, I have provided English translations for examples, especially in qualitative analysis and tables (e.g., Table 4 and Table 5).
I have included a section discussing the limitations of the research, potential threats to validity, and the practical implications of the study on both research and real-world applications.